Deriving fibroblast cell lines from wing-punch biopsies of Australian eastern bent-winged bats (Miniopterus orianae oceanensis)

Langguth Anna anna03.langguth@gmail.com 1
Brannelly Laura A. 1
Turbill Christopher 2 3
Villada-Cadavid Tomás 3
Wu Nicholas C. 3 4 5
Hufschmid Jasmin 1
Cottingham Ellen 6
1 One Health Research Group, Melbourne Veterinary School, University of Melbourne , Melbourne , Victoria , Australia
2 School of Science, University of Western Sydney , Richmond , New South Wales , Australia
3 Hawkesbury Institute for the Environment, University of Western Sydney , Richmond , New South Wales , Australia
4 Centre for Terrestrial Ecosystem Science and Sustainability, Harry Butler Institute, Murdoch University , Murdoch , Western Australia , Australia
5 School of Environmental and Conservation Science, Murdoch University , Murdoch , Western Australia , Australia
6 The Asia Pacific Centre for Animal Health, Melbourne Veterinary School, University of Melbourne , Melbourne , Victoria , Australia
Brygadyrenko Viktor
Electronic publication date: 2025 Oct 23
Publication date: 2025
Volume: 13
Electronic Location ID: e20222
Received 2025 Feb 27; Accepted 2025 Sep 21
Copyright: ©2025 Langguth et al.
Copyright year: 2025
Copyright holder: Langguth et al.
License: This is an open access article distributed under the terms of the Creative Commons Attribution License, which permits unrestricted use, distribution, reproduction and adaptation in any medium and for any purpose provided that it is properly attributed. For attribution, the original author(s), title, publication source (PeerJ) and either DOI or URL of the article must be cited.
License URL: https://creativecommons.org/licenses/by/4.0/

Keywords: Cell culture, Fibroblasts, Eastern bent-winged bat, Non-model organism, Wing-punch biopsies, In vitro models

Funding: Australian Research Council Linkage Project LP2000100331 The Holsworth Wildlife Research Endowment Fund, administered by the Ecological Society of Australia Anna Langguth was supported through an Australian Research Council Linkage Project grant (LP2000100331). This research was also funded by the Holsworth Wildlife Research Endowment Fund, administered by the Ecological Society of Australia. The funders had no role in study design, data collection and analysis, decision to publish, or preparation of the manuscript.

==============================
Cell cultures are a valuable tool for the study of in vitro disease processes, especially where such processes concern wild and/or threatened animal species. However, the collection of adequate samples for cell line preparation can be challenging under field conditions due to logistical challenges and access to equipment. In this paper, we describe the generation of fibroblast cell lines derived from wing-punch biopsies of Australian eastern bent-winged bats (Miniopterus orianae oceanensis), expanding on and modifying existing protocols. Twenty wing-punch biopsies were collected from free-ranging individuals in New South Wales in February 2024 and shipped to the University of Melbourne, Victoria, within 24 hours. To assess the impact of different preservation methods on sample integrity, samples were subjected to two different shipping treatments: Ten were snap-frozen immediately upon collection, and the other ten were placed in cool phosphate-buffered saline (PBS) for transport. To test the effect of different plating treatments, samples were plated as either collagenase-digested cells or explants. Although none of the frozen biopsies or explants showed any growth, all samples transported in cool PBS and plated as digested cells remained viable. While agitation of the samples prior to plating resulted in an initially faster rate of cell growth, cells derived from tissue that had been digested multiple times spread across the plate and formed a monolayer significantly faster than those that had been digested only once. This study confirms the effectiveness of existing cell culture protocols using non-lethal sampling techniques in an Australian insectivorous bat species and shows a novel method of maximizing cell yield from a single biopsy. It also introduces an alternative transportation method, beneficial for field sample collection. These cell cultures are essential tools for future studies on disease susceptibility and pathogen responses in bat species, particularly those belonging to the family Miniopteridae. Additionally, they can be used for biobanking efforts, preserving the genetic material of non-model organisms for broader conservation purposes.

Introduction

In recent years, studies using isolated tissues and cells have emerged as a valuable alternative to live animal trials in wildlife health research (Gultom et al., 2021; Kohl et al., 2021; Alcock et al., 2024). Cell cultures are an especially promising method for the simulation of disease progression across various organs, allowing an in-depth understanding of underlying mechanisms of pathogenicity and host response. As such, they have been used to study disease susceptibility and dynamics in a number of wild animals (e.g., Gultom et al., 2021; Essbauer et al., 2011; Pham et al., 2020), including bats (Wynne et al., 2016; Aurine et al., 2021).

Mammalian cell culture also serves as a source of genetic material from species of special conservation importance. Researchers may preserve genetic material for biobanking purposes, defined as the long-term storage and preservation of biological samples for future research, conservation, or potential reintroduction efforts (Mooney et al., 2023) but may also use the material to investigate cellular responses to stressors where characterization in live animal models may not be feasible (Webb et al., 2014). Beyond genetic characterization, these cultures hold promise for reproductive technologies such as induced pluripotent stem cell derivation (Wu et al., 2024), where easily obtainable somatic cells such as skin fibroblasts can be reprogrammed into other cell types, including germ cells or even gametes, to support species recovery efforts (Mattos et al., 2023). Additionally, cultured cells act as living repositories for future functional studies, as they can be thawed and analyzed as new molecular tools emerge. For example, some advanced techniques currently used in human medicine now allow simultaneous measurement of genetic activity and surface protein markers at the single-cell level (Liu et al., 2023). If adapted to biobanked tissues, this approach would offer a powerful system to dissect immune responses to disease stimuli such as white-nose syndrome in bats.

Bats are the only mammals capable of self-powered flight (Teeling, 2009). This capability seems to have awarded them several evolutionary advantages, such as greater longevity relative to other small mammals (Podlutsky et al., 2005). Additionally, flight has likely contributed to the evolution of unique immune responses in bats, as their ability to rapidly move between diverse environments and therefore host a wide range of pathogens has driven the development of immune adaptations that allow them to carry pathogens without necessarily showing clinical signs of illness themselves (Banerjee et al., 2020; O’Shea et al., 2016). The immune system of bats has garnered substantial research interest, as it could have implications for both bat and human health as well as disease management (Wibbelt et al., 2010). Bat cell cultures have been used extensively to investigate the viral diversity hosted by a diverse number of bat species (Kohl et al., 2021), immunological processes (Wynne et al., 2016; Banerjee et al., 2017), susceptibility to diseases such as the zoonotic Hendra virus (Wynne et al., 2016; Aurine et al., 2021; Crameri et al., 2009), and bat-specific pathogens such as Pseudogymnoascus destructans, which causes the potentially fatal white-nose syndrome (Isidoro-Ayza & Klein, 2024; Cornelison et al., 2014).

For cell culture approaches to be practical in a conservation context, the tissue samples used must be easy to obtain, cause no lasting harm to the animals they are collected from, and should be collected through measures that are quick and easy to implement without posing a health hazard to field personnel or the animals involved. Wing-punch biopsies are a safe, minimally invasive way to obtain bat skin tissue for cell derivation, with a standard 3-millimeter (mm) biopsy wound healing completely within 16 days (Weaver et al., 2009).

Previous work in which fibroblast cell lines were derived from bat wing tissue has encountered several difficulties, including low cell yield (Alcock et al., 2024; Kacprzyk et al., 2016), attachment difficulties (Cornelison et al., 2014), and extended culture times (Yohe et al., 2019). Explant cultures often require long incubation periods—sometimes over 30 days—before observable cell growth occurs (Wynne et al., 2016). In these explant-based methods, tissue adherence to culture plastic has also required external supports such as meshes (Cornelison et al., 2014). Additionally, digestion protocols used in some prior studies have typically relied on one incubation only, using a single enzyme (Kacprzyk et al., 2016; Yohe et al., 2019; Field et al., 2015), which may be insufficient to fully dissociate skin samples.

In this paper, we expand upon and optimize an existing protocol to generate bat wing fibroblasts from wing-punch biopsies published by Yohe et al. (2019) and describe an alternate method of tissue transportation as well as a method for optimizing cell yield.

Materials & Methods

Field sample collection

Following previous studies in which wing-punch biopsies of five individuals per group were compared in terms of cellular responses to a stimulus (Field et al., 2015), we decided to allocate five wing-punches to each of our four initial treatments (frozen, plated as explant; frozen, plated as digested cells; cooled, plated as explant; cooled, plated as digested cells). Therefore, 20 wing-punch biopsies were collected, one from each of 20 free-ranging eastern bent-winged bats (Miniopterus orianae oceanensis) at a cave in the Jenolan Karst Conservation Reserve in February 2024. Our methodology followed established protocols working with free-ranging Australian bats (Lumsden et al., 2022), as well as wildlife capture and marking recommendations (Waudby et al., 2022b; Waudby et al., 2022a). All biopsies were collected from healthy adult bats without any prior wing damage. Animals were trapped as part of other ongoing studies (Western Sydney University, Animal Care and Ethics Committee; A14849; New South Wales Department of Climate Change, Energy, the Environment and Water; SL 102671) using a harp trap monitored continuously in front of the cave entrance (Lumsden et al., 2022). Individuals were aged by examining wing joint fusion (Parnaby, 1999). Bats were then weighed to the nearest 0.1 g with a digital scale (MS500; PESOLA Präzisionswaagen AG, Schindellegi, Switzerland) and had their right forearm (carpus to elbow) measured to the nearest 0.1 mm with digital calipers (B09QL1RB1G; Ziyuan Electronics Co., Ltd., Guangdong, China). Their sex was then determined by examining external genitalia (n = 10 males and n = 10 females).

In accordance with standard field techniques, bats were only restrained for less than a minute to minimize stress associated with handling (Corthals et al., 2015). To collect wing-punch biopsies, the animal’s wing was gently extended and both sides of the uropatagium were disinfected with 80% v/v ethanol. Samples were collected with a sterile three mm biopsy punch (Miltex biopsy punch, T984-30; ProSciTech, Thuringowa, QLD, Australia), avoiding blood vessels, which are visible when the wing is extended. All bats were released after sample collection.

Ten biopsies were transferred into individual 4.5 mL CryoVials (151.303.03P; Pacific Laboratory Products, Blackburn, Australia), each containing 2 mL of cryopreservative medium prepared according to Son et al. (2021), with the addition of 20% v/v FBS (#10082147; Thermo Fisher Scientific, Waltham, MA, USA). These biopsies were transferred into liquid nitrogen within three minutes after collection (“frozen” biopsies).

Ten additional biopsies were directly transferred into individual 15 mL Falcon tubes (100-0092; StemCell Technologies, Vancouver, BC, Canada), each containing 2 mL cold Phosphate-buffered saline (PBS) with 1% v/v Antibiotic-Antimycotic (10,000 units/mL Penicillin, 10,000 µg/mL Streptomycin, 25 µg/mL Amphotericin B; 15240062, Thermo Fisher Scientific, Waltham, MA, USA; “cooled” biopsies). All biopsies were transported to Melbourne within 19 h after collection, with snap-frozen samples kept on dry ice and those in cool PBS transported in a portable cooler at 4 °C. Upon arrival at the laboratory, 24 h after collection, the “cooled” samples were transferred into 2 mL of culture medium maintained at 4 °C.

Laboratory methods

Cell cultures were grown to test variations on a protocol previously described by Yohe et al. (2019). Our aim was to improve the success rate of culturing cells from wing punch biopsies, while simplifying sample collection. Our modifications included sample transport in PBS, attempting to grow cells from explants, and successive digestion steps of sampled tissue.

Prior to adding cells, we coated all wells of a 6-well tissue culture plate (#140675; Thermo Fisher Scientific, Waltham, MA, USA) using 1 mL 2% w/v gelatine (G1393; Sigma-Aldrich, St. Louis, MO, USA) per well to improve cell adherence. We then added 2 mL of fresh, pre-warmed growth medium (37 °C) to each well. The growth medium consisted of Dulbecco’s Modified Eagle Medium (#36256; StemCell Technologies, Vancouver, BC, Canada) supplemented with 20% v/v FBS (#10082147; Thermo Fisher Scientific, Waltham, MA, USA), 50 µg/mL Gentamycin (#15710064; Thermo Fisher Scientific, Waltham, MA, USA), and 1% v/v Penicillin/Streptomycin (#SV30010; HyClone, Logan, UT, USA). One well was designated for each of the wing-punch biopsies collected. Plates were covered with labelled lids and stored in a 37 °C incubator with 5% CO2 until needed.

Our modified protocol involved plating half of the samples as explants (n = 10). Five “frozen” and five “cooled” biopsies were plated as explants 40 h after collection (Table 1). “Frozen” samples were placed in a warm water bath at 37 °C for a maximum of three minutes to rapidly thaw them. Each biopsy was then rinsed in one ml of PBS and plated in the middle of the well, to which 1 mL of pre-warmed (37 °C) culture medium was added. To attempt explant plating, we first placed the biopsies into the center of the wells, allowing them to adhere in 1 mL of medium for a short period. To promote attachment, we also gently scoured the explants into the base of the wells. For digested cells (n = 10), our modified protocol involved testing two successive digestion steps alongside a single digestion.

Table 1 Wing punch biopsy samples used to compare effects of frozen (−20 °C) versus cooled (4 °C) tissue transport, and to evaluate cell growth from samples plated either as digested cells or explants.

Samples were collected from 20 adult eastern bent-winged bats (Miniopterus orianae oceanensis) trapped in the Jenolan Karst Conservation Reserve in February 2024.

Bat ID	Sex	Transport method	Plating method	
29	Female	Frozen (−20 °C)	Explant	
30	Female	Digested	
31	Female	Digested	
32	Male	Explant	
33	Female	Explant	
34	Male	Digested	
35	Male	Explant	
36	Male	Digested	
37	Male	Digested	
38	Female	Explant	
39	Female	Cooled PBS (4 °C)	Explant	
40	Female	Digested	
41	Female	Digested	
42	Female	Digested	
43	Male	Digested	
44	Male	Explant	
45	Female	Digested	
46	Male	Explant	
47	Male	Explant	
48	Male	Explant	

In addition, five “frozen” and five “cooled” biopsies were digested to be plated as digested cells (Table 1). Again, “frozen” biopsies were rapidly thawed in a warm water bath as described above and were then rinsed in 1 mL PBS. Wing-punch biopsies were first digested overnight at 37 °C using collagenase IV (one mg/mL; 07909, StemCell Technologies, Vancouver, BC, Canada), starting 45 h after collection. The digestion was halted after 15 h by adding fresh, pre-warmed (37 °C) growth medium, in which the cells were re-suspended. Undigested tissue was removed using flame-sterilized forceps and transferred to a separate Eppendorf tube (#00301238301; Eppendorf, Hamburg, Germany).

Digested cells were then prepared for plating by centrifuging for 3 min at 300 ×g (Westlab Smart Micro Centrifuge, 663-958; Westlab, Ballarat, VIC, Australia). The supernatant was discarded, and the cell pellet re-suspended in 500 µL of warm (37 °C) growth medium. The resulting cell suspension for each sample was then each added to one well of the 6-well tissue culture plate (Digested cells, Subset 1). The undigested tissue was mixed with 500 µL of warm (37 °C) growth medium and gently agitated by repeated passage through a pipette to fragment it into smaller components. The resulting suspension was plated by adding it to one well of the 6-well tissue culture plate (Digested cells, Subset 2). Any remaining wing tissue that had not dissolved was subjected to a second digestion with Collagenase IV and incubated overnight for an additional 15 h. After this second digestion, the suspension was centrifuged, re-suspended, and plated as described (Digested cells, Subset 3).

The medium in each well was changed every 12 h during the initial five days of the study, followed by a change every 48 h for the remainder of the 20-day study period. Cell growth was observed via light microscopy (Olympus BX, Tokyo, Japan) at least once every 24 h, recording the percentage of coverage on the bottom of each well.

Confluent wells (cells covering ≥ 80% of the bottom of the well) were passaged as described previously (Field et al., 2015). To assess the feasibility of permanent cell-line storage using liquid nitrogen, a subset of confluent samples was frozen down after 12 days of growth. To do this, cells were passaged, and half of the cells contained in the original well were centrifuged at 300 ×g for three minutes. The resulting pellet was then resuspended in 1 mL freezing medium, which consisted of culture medium supplemented with 10% v/v dimethyl sulfoxide (DMSO) (#HB3262; HelleBio, Bristol, UK). The cell suspension was stored within a CryoVial (#664-982; Mitchell Park, VIC, Australia). Vials were then placed into a cryogenic freezing container (Corning CoolCell LX; Bio-Strategy Pty Ltd, Melbourne, VIC, Australia) and were immediately transferred into a −80 °C freezer. After 27 h, samples were transferred into liquid nitrogen. Cells were revived from liquid nitrogen by gently adding 4 °C culture medium to cryopreserved cells in CryoVial in a dropwise manner while gently flicking the tube. Once defrosted, the cells were added to approximately 14 mL of 4 °C culture medium to rapidly dilute out the cryoprotectant. The cells were then centrifuged at 700 g for 5 min. The supernatant was removed, and cells were resuspended gently in culture medium after which they were plated onto a 24-well plate. Cells were observed 24 h later via light microscopy (Olympus CKX53; Olympus, Tokyo, Japan).

Statistical analysis

Analysis of factors influencing cell growth was carried out using R software version 04.2.2 (R Core Team, 2020), focusing only on samples in which growth was observed. We used the ‘lme4’ (Bates et al., 2015) package (version 1.1-28) to run linear mixed-effects (LME) models assessing how the number of digestion steps (‘1’ or ‘2’) and sample agitation (‘yes’ or ‘no’) influenced two outcome variables: time to first growth and time to confluence (both measured in days). These predictors were included as fixed effects, with separate models constructed for each response variable. The random effect included ‘Bat ID’ to account for the repeated measures, as each fresh, digested wing-punch biopsy from an individual bat (n = 5) was processed into three cell culture subsets. We assessed model assumptions by visually inspecting Q–Q plots of residuals for normality, and residuals versus fitted value plots for homoscedasticity and linearity. These diagnostics confirmed that assumptions were not violated. Models for both outcome variables were then further evaluated using the ‘car’ package to perform an ANOVA (Fox & Weisberg, 2019).

Given the relatively low sample size per experimental condition, post hoc power analyses were conducted for both primary response variables using the ‘pwr’ package (Champely, 2020) (version 1.3.-0) to assess whether the study design was sufficient to detect meaningful differences.

Results

We observed no growth of cells derived from any biopsies that were snap-frozen in the field or plated as explants without a digestion step. When cooled biopsies were digested, four out of five samples showed fibroblast growth. No bacterial contamination was observed in any of the wells throughout the course of the 20-day experiment.

Survival of cells from permanent storage in liquid nitrogen resulted in considerable cell death, with only approximately 10–20% of cells surviving and adhering to the plate. However, surviving cells recovered well and continued to proliferate with a population doubling rate of approximately 48 h (Fig. 1).

Power analysis with a sample size of five per treatment group indicated that biologically meaningful differences between treatments could be detected, with minimum detectable proportional changes of approximately 3.6% for time to confluence and 49.4% for time to first growth (α = 0.05, power = 0.8).

For cooled cells that had been digested only once (Digested cells, Subset 1, n = 4), no cell growth was observed until after two days. Fibroblast growth in ‘Digested cells, Subset 2’ (digested and agitated, n = 4) and ‘Digested cells, Subset 3’ (digested twice and agitated, n = 4) was observed as early as one day after plating. Confluence was first reached in ‘Digested cells, Subset 3’, with the earliest confluent well being observed five days after plating (Table 2).

Figure 1 Primary cell line derived from wing-punch biopsies of Australian eastern bent-winged bats (Miniopterus orianae oceanensis) demonstrating fibroblast morphology.

(A) Brightfield microscope image of fibroblast cells at approximately 20% confluency. (B) Brightfield microscope image of Fibroblast cells at near 100% confluency. Scale bar 200 µm.

The timing of first observed cell growth was significantly affected by whether cells were agitated before they were plated (LME; Agitation: X2 = 5.65, df = 1, p = 0.02). First growth in agitated samples was seen about one day earlier than in non-agitated samples (Fig. 2, Table 2).

In contrast, the time it took for each cell culture plate to reach confluence was significantly affected by the number of digestion steps before plating (LME; Digestion: X2 = 29.86, df = 1, p < 0.01).

Cells derived from samples digested twice reached confluence significantly faster (median = 11.0 days (IQR: 1.5)) than those digested once (median = 16.3 days (IQR: 0.33 days)), representing a 32.5% reduction in time to confluence (Fig. 3, Table 2).

Table 2 Median time plus interquartile range (IQR) in days for time until first growth and confluence of fibroblast cells after three different treatments.

Cells were derived from wing punch biopsy samples of eastern bent-winged bats (Miniopterus orianae oceanensis, n = 4) trapped in the Jenolan Karst Conservation Reserve in February 2024. Differences between treatments were assessed using linear mixed effects models, with Bat ID as a random factor to account for repeated measures within individuals.

Subset	Explanation	First growth (days)	Confluence (days)	
		Median	IQR	Median	IQR	
1	Digested cells, once	2.25	0.63	16.30	0.33	
2	Digested cells, once + agitation	1.50	1.00	15.00	1.08	
3	Digested cells, twice	2.00	0.25	11.00	1.50	

Figure 2 Time (in days) until first fibroblast growth was observed in three subsets of cell cultures derived from wing punch biopsies of eastern bent-winged bats (Miniopterus orianae oceanensis,n = 4).

All samples were transported in chilled PBS and digested 45 h after collection. Each point represents a biological replicate from one bat. All treatments were performed on four wing punch biopsies, with cells plated after each processing step as replicates from the same original tissue. Red points indicate median values, with error bars representing the interquartile range. Asterisks indicate significance levels (p ≤ 0.05: *).

Figure 3 Time (in days) until confluence was first observed in three fibroblast culture subsets derived from wing punch biopsies of eastern bent-winged bats (Miniopterus orianae oceanensis, n = 4).

All samples were transported in chilled PBS and digested 45 h after collection. Each point represents a biological replicate from one bat. All treatments were performed on four wing punch biopsies, with cells plated after each processing step as replicates from the same original tissue. Red points represent median values, with error bars indicating the interquartile range. Asterisks indicate significance levels (p ≤ 0.05: *, p ≤ 0.01: **).

Discussion

Working with free-ranging bats and collecting tissue samples such as wing-punch biopsies in the field presents several challenges. Capturing bats from difficult-to-access locations can limit sample sizes (Turbill, 2006; Campbell et al., 2022), making it crucial to maximize the quality and utility of each collected sample. Additionally, access to liquid nitrogen may not always be available in all settings.

Our results validate the use of a previously published protocol (Yohe et al., 2019) for efficiently deriving cell lines from Australian bat wing-punch biopsies. Our work also demonstrates that chilled PBS is a simple yet effective medium for the short-term transport of tissue samples. Furthermore, our findings highlight the benefits of multiple digestion steps for achieving a higher cell yield from potentially limited samples.

We found that using sterile PBS with 1% v/v Antibiotic-Antimycotic is an effective medium for transporting wing-punch biopsies. Although this medium contains fewer nutrients than standard cell transport media (Yohe et al., 2019), cells remained viable for at least 24 h. Although skin biopsies are minimally invasive compared to biopsies from internal organs (Isidoro-Ayza & Klein, 2024), bat wing tissue contains diverse and abundant commensal flora (Holz et al., 2018; Johnson et al., 2013) and cannot be collected in a completely sterile manner. This raises concerns for bacterial over-colonization and potential deterioration of samples during transport (Stacey, 2011). Transporting wing-punch biopsies in nutrient-poor, cold PBS using a portable cooler presents a cost-effective option for maintaining the viability of the samples and preventing bacterial overgrowth during short-term transport to the laboratory, without the need for using hazardous substances such as liquid nitrogen.

While snap-freezing samples after collection is the gold-standard to preserve tissue integrity if long-term storage is required, none of our frozen samples were viable. At a temperature of −196 °C (Isac-García et al., 2016), liquid nitrogen instantly freezes tissue, reducing ice crystal formation that can compromise cell integrity and viability in cell cultures (Meryman, 1956). The use of a cryopreservative additionally mitigates ice crystal formation by lowering the melting point of free water (Wowk, 2007). Although we ensured appropriate contact time between the cryoprotectant and wing tissue before snap-freezing, we were unable to derive viable cells from these samples. One potential explanation is the ‘Leidenfrost effect’ (Song et al., 2010), which has been described in both human (Rivas Leonel, Lucci & Amorim, 2019) and animal (Imrali et al., 2020) cryopreservation studies. Because cryovials were not pre-chilled, the extreme temperature difference upon immersion in liquid nitrogen likely caused a vapor barrier to form around the vials, temporarily insulating the contents and slowing the freezing process (Imrali et al., 2020). Future work should consider pre-chilling cryovials to mitigate this issue.

It would also be useful to test different cryoprotectants and identify which stage of the process—freezing, thawing, or cryoprotectant exposure—may have led to cell damage. The cryopreservative we chose in our study, DMSO, is a commonly used reagent that has been successfully applied for the long-term storage of a wide range of exotic animal cell lines (Mattos et al., 2023; Borges et al., 2020; Siengdee et al., 2018). However, future research may wish to trial alternative cryopreservation strategies (Elliott, Wang & Fuller, 2017; Lazareva et al., 2022). Species-specific differences in cryoprotectant efficacy have been demonstrated; for example, fibroblasts from three closely related felid species required varying concentrations of DMSO for optimal preservation (Arantes et al., 2021). One promising direction for future work involves replicating physiological adaptations observed in torpid bats during cooling (Nemcova et al., 2023): The addition of high concentrations of glucose exerted a cryoprotective effect on bat kidney, liver, and nerve cells, as well as on macrophages. Other studies have used combinations of DMSO and sucrose for successful fibroblast cryopreservation (Borges et al., 2020). Cryoprotectants can be toxic to cells if exposure is prolonged, so further strategies to rapidly dilute the cryoprotectant upon removal from liquid nitrogen should be explored. Furthermore, given the thin nature of the biopsies, even the minimal exposure before snap-freezing might have been enough to cause cell damage. This warrants further investigation into the optimal timing for wing-punch biopsy freezing and thawing.

It is further worth noting that when tissue is frozen slowly, ice crystal formation is mainly observed in extracellular spaces (Chambers, Hale & Hardy, 1932; Meryman & Platt, 1955; Müller-Thurgau, 1886). To address this, recent work published after our samples were collected (Deng et al., 2024) proposed an alternative preservation method: enzymatically digesting bat wing tissue prior to cryopreservation, followed by slow freezing in cryoprotectant within a cryoprotective container that enables a controlled cooling rate of approximately 1 °C per minute. This technique aligns with established protocols for long-term cell storage (Field et al., 2015) and may reduce extracellular ice formation by breaking down structural tissue prior to freezing. While this method would require validation in other bat species, it may offer a more reliable alternative for future work, especially when immediate cell culture is not feasible.

We did not observe growth in any of our explants. Few studies specifically focus on bat wing explants, and only one study has used bat wing explants to derive fibroblasts (Aurine et al., 2021). One study used bat wing cell explants to examine how bacterial volatile agents inhibit cellular invasion by Pseudogymnoascus destructans in pathogen-inoculated tissue (Cornelison et al., 2014); however, explants had to be adhered to a mesh support to ensure they stayed in place. This indicates that cell growth did not occur, as the use of the mesh would not have been necessary if cells had grown out of the explants and attached themselves to the plate surface. However, the primary goal of this work was simply to maintain the tissue in a culture medium, not derive cells from it.

In our study, although we used a culture medium suitable for promoting cell growth, our explants might also have had difficulties attaching to the plates. Despite coating the wells with gelatine, the microscopic hairs on the explants might have hindered cells from reaching and growing into the gelatine layer. This issue has been noted in lab mouse studies, where tissue biopsy samples to be used as explants are instead collected from the subcutaneous layer, largely avoiding skin hair. Additionally, contact between the explant and the culture surface is sometimes facilitated by sandwiching the tissue with a coverslip, secured to the plate using silicone grease (Son et al., 2024).

The reason why we were unable to derive fibroblasts from our explants may also simply be a matter of timing. The only previous study in which fibroblasts were derived from bat wing tissue explants similarly reported slow growth (Aurine et al., 2021). Cells derived from wing-punch biopsies grew significantly slower than those obtained from visceral tissue explants, with first cells only emerging after 30 days. It is therefore possible that we would have observed cell growth in plated explants if these samples had been maintained continuously for longer than the 20 days of our experiment. However, our results indicate that deriving cells from digested tissue is significantly faster and therefore a more cost-effective method for rapidly generating primary fibroblast cell lines.

Digestion of bat wing skin tissue twice instead of once markedly improved overall cell growth, and cells took significantly less time to reach confluence. Recent work on human skin samples further supports our findings. A 2023 study reported that cell yield per cm2 of skin was 3.4 times higher with double digestion compared to a single digestion (Sierra-Sanchez et al., 2023). Another study described a protocol combining one mechanical and two digestion steps, which increased cell counts by 28% (Burja et al., 2022).

While no other studies have assessed the effect of multiple digestion steps on bat wing skin tissue, findings in other insectivorous bat species also underpin the fact that single digestion leads to slow cell growth. Wing-punch biopsies of lesser horseshoe bats (Rhinolophus hipposideros) that underwent only one digestion step reached confluence only after approximately three weeks, despite the fact that cell cultures were supplemented with additional growth factors and L-Glutamine (Kacprzyk et al., 2016). This prolonged time to confluence seems to be a likely consequence of the digestion method, which then resulted in fewer single cells that would proliferate on their own (Burja et al., 2022).

We chose collagenase type IV for tissue digestion due to its milder activity (Woodley et al., 1986), appropriate for the relatively thin wing punch biopsies used in this study. Our observed growth rates to confluence (∼11–16 days) are comparable to those reported in other mammals using similar biopsy techniques, such as Asian elephants (Elephas maximus, 12–13 days) (Siengdee et al., 2018) and manatees (Trichechus inunguis, ∼14 days) (Tavares et al., 2024). Other enzymatic methods, including trypsin (Wang et al., 2020; Hashem et al., 2007; Han et al., 2001; Abade Dos Santos et al., 2021), and combinations with mechanical dissociation (Jansen Van Vuuren et al., 2023), have been successfully applied in various species. However, trypsin is not recommended for bat wing samples due to the tissue’s high elastin content, which trypsin does not effectively digest (Holbrook & Odland, 1978). Further research is needed to optimize digestion protocols specifically for fibroblast isolation from insectivorous bat wing-punch biopsies to improve cell yield and growth.

The sample size for this study was limited (samples from 20 individuals, with 5 per treatment group); therefore, the inferences drawn—particularly regarding the timing of initial growth—might be limited. Nonetheless, we detected a statistically significant effect of agitation, with agitated samples achieving growth approximately one day earlier than non-agitated ones. These preliminary findings provide biologically meaningful evidence supporting the use of agitation in combination with tissue digestion. Future studies with larger sample sizes will be valuable for refining our understanding of early fibroblast outgrowth and for further characterizing inter-individual variability in growth dynamics.

Overall, the current sample size provided sufficient statistical power for detecting differences in our primary endpoint (confluence), while also providing valuable preliminary insights into early growth patterns to inform future work.

Conclusions

In conclusion, we were able to confirm the efficacy of an existing cell culture protocol (Field et al., 2015) for use in Australian insectivorous bats, although we recommend adding a second tissue digestion step for optimized cell yield and growth rates. Our study also demonstrates that cold PBS is a viable alternative to cell culture medium for tissue transport, potentially reducing the risk of bacterial overgrowth. However, a limitation of this approach is the need to expedite tissue transport to preserve sample viability. While the maximum duration that samples can remain in PBS before being placed in culture medium was not tested, minimizing this time (ideally to under 24 h) is likely beneficial for cell viability. These optimizations will be useful in cases where minimal sample is available for cell culture work. Successfully deriving cell lines from non-model organisms such as bats, especially endangered species, not only ensures that genetic material is available for future research but also plays a crucial role in long-term conservation efforts. This approach enhances our ability to study disease dynamics in controlled settings, facilitating future conservation and health monitoring of Australian bats.

Supplemental Information

Supplemental Information 1 Raw data on growth of cells derived from wing punch biopsies of Australian eastern bent-winged bats (Miniopterus orianae oceanensis)

Growth data from all explants collected and grown during the course of the experiment. This data was used for statistical analysis to compare the effects of different storage, plating, and processing treatments.

Supplemental Information 2 ARRIVE 2.0 Checklist

We would like to express deep gratitude to Denise O’Rourke for her technical advice. We would also like to thank Andrew Gherlenda and Ronald Walsh for their help with sourcing liquid nitrogen shippers and permanent storage units.

Additional Information and Declarations

Competing Interests

Author Contributions

Animal Ethics

Field Study Permissions

Data Availability

Laura A. Brannelly is an Academic Editor for PeerJ.

Anna Langguth conceived and designed the experiments, performed the experiments, analyzed the data, prepared figures and/or tables, authored or reviewed drafts of the article, and approved the final draft.

Laura A Brannelly conceived and designed the experiments, analyzed the data, authored or reviewed drafts of the article, funding acquisition, Project administration, and approved the final draft.

Christopher Turbill conceived and designed the experiments, authored or reviewed drafts of the article, funding acquisition, Fieldwork resources, and approved the final draft.

Tomás Villada-Cadavid performed the experiments, authored or reviewed drafts of the article, fieldwork resources, and approved the final draft.

Nicholas C Wu analyzed the data, authored or reviewed drafts of the article, fieldwork resources, and approved the final draft.

Jasmin Hufschmid conceived and designed the experiments, authored or reviewed drafts of the article, funding acquisition, Project administration, and approved the final draft.

Ellen Cottingham performed the experiments, authored or reviewed drafts of the article, contributed reagents, materials, general advice on cell culture work, and approved the final draft.

The following information was supplied relating to ethical approvals (i.e., approving body and any reference numbers):

Samples were collected as part of a study approved by Western Sydney University’s Animal Care and Ethics Committee (WSU ACEC). Approval number: A14849.

The following information was supplied relating to field study approvals (i.e., approving body and any reference numbers):

Field research was approved by the WSU ACEC, approval number: A14849 and the NSW Department of Planning and Environment (NSW DPIE), approval number: SL102671.

The following information was supplied regarding data availability:

The raw data is available in the Supplemental File.

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
