# Peer review of "Deriving fibroblast cell lines from wing-punch biopsies of Australian eastern bent-winged bats (Miniopterus orianae oceanensis)"

_PeerJ, doi:10.7717/peerj.20222_

## Round 0.1 · original submission · Minor Revisions

This manuscript optimizes the existing protocol of deriving fibroblast cell lines from wing-punch biopsies of Australian eastern bent-winged bats, and thus potentially benefits future studies. Yet, some minor additions are suggested prior to publication, including (1) proposing alternative digestion and other conditions (it might be helpful to have a quick table summarizing what have been studied/published and what are considered by the authors as suitalbe future optimizations), (2) discussing how this improved protocol would be transferable to another species, and (3) providing some details about the figures, as pointed out by the reviewers.

Reviewer 1 ·

Basic reporting

The manuscript is well-written in clear and professional English. The introduction provides a strong background on the relevance of fibroblast cell culture in wildlife research, particularly for bats. The literature cited is appropriate and provides sufficient context to justify the study. The figures and tables are well-structured, with clear labels and descriptions that effectively support the findings. The raw data has been provided and appears to align with the results presented in the manuscript.

Overall, this is a well-conducted study that provides practical insights into fibroblast cell culture from bats. The manuscript contributes useful information to the field of wildlife disease research and conservation biology. Addressing the points raised above will further strengthen the study.

Experimental design

The study is within the scope of PeerJ and presents original research on an important topic related to conservation and wildlife disease research. The research question is well-defined, and the methodology is rigorous and detailed, ensuring reproducibility. The experimental design comparing different transport and plating methods is well thought out and addresses practical challenges in field-based sample collection.

However, I have the following concerns regarding the experimental methods:

1. Cell Viability in Cryopreserved Samples – The failure of fibroblast growth in frozen samples is noted, and the authors suggest that the "Leidenfrost effect" might have played a role. It would strengthen the manuscript if the authors could discuss alternative strategies for optimizing cryopreservation further. For example, were different cryoprotectant formulations considered?
2. Sample Size Consideration – While the study includes a reasonable number of biopsies (n=20), the number of samples per experimental condition is relatively low (n=5 per group). The statistical analyses are well presented, but a discussion of potential variability in cell growth rates between individual animals would be beneficial.
3. Digestion Protocol Optimization – The finding that multiple digestions improve cell yield is significant. However, it would be useful to provide additional discussion on whether different enzyme concentrations or alternative digestion strategies (e.g., mechanical disruption) were tested in prior literature.
4. Need to cite more recent novel papers in discussion section for future studies: 1. High-plex protein and whole transcriptome co-mapping at cellular resolution with spatial CITE-seq. 2.Spatial dynamics of mammalian brain development and neuroinflammation by multimodal tri-omics mapping. 3. Spatially resolved in vivo CRISPR screen sequencing via perturb-DBiT.

Validity of the findings

The results are clearly presented, and the findings provide valuable insights into fibroblast culture from bat wing biopsies. The authors effectively demonstrate that chilled PBS transport is a viable alternative to liquid nitrogen, which has significant implications for field studies.

The statistical analyses support the conclusions drawn. However, some additional details could improve the discussion:

Comparison to Other Species – The manuscript discusses previous bat studies but could benefit from comparisons to other mammalian fibroblast culture methods. Are the cell growth rates observed here comparable to fibroblast cultures from other species using similar biopsy techniques?
Implications for Biobanking – While the authors briefly mention biobanking, a more detailed discussion on how these fibroblast lines could be preserved for long-term conservation efforts would be valuable.

Reviewer 2 ·

Basic reporting

no comment

Experimental design

no comment

Validity of the findings

no comment

Additional comments

Here are some of the concerns that the author may want to address, for being finally accepted to be published.

<1> Figure 1 and Figure 2 quality does not seems to meet the general standards. Please redo them in a more clear, and more appealing way. Increasing letter font size, adding notes on data and experimental design. For example, numbers of biological and technique replications, does each dot represent a biological replication or otherwise? The significance level represented by the asterisk?

<2> Brief statistics and experimental design are also expected for Table 1 and Table 2.

Reviewer 3 ·

Basic reporting

The literature is well-referenced and supports the study rationale. However, further discussion on previous challenges in bat fibroblast culture protocols could enhance the background.
Figures and tables are well-structured and support the findings. Figure captions should be more descriptive to aid standalone interpretation.

Experimental design

Please clarify if the sample size of 20 wing-punch biopsies can provides sufficient statistical power.

Validity of the findings

The statistical analyses (LME models, ANOVA) are appropriate. However, the model assumptions and validation procedures (e.g., residual analysis) should be explicitly mentioned. And the discussion on why explants failed to generate viable cultures could be expanded by comparing with similar studies.

---

## Round 0.2 · accepted · Accept

Dear Dr. Langguth, I congratulate you on the acceptance of this article for publication.

Reviewer 1 ·

Basic reporting

-

Experimental design

-

Validity of the findings

-

Reviewer 3 ·

Basic reporting

-

Experimental design

-

Validity of the findings

-